# TABKANET: TABULAR DATA MODELING WITH KOLMOGOROV-ARNOLD NETWORK AND TRANSFORMER

## ABSTRACT

Tabular data is the most common type of data in real-life scenarios. In this study, we propose the TabKANet model for tabular data modeling, which targets the bottlenecks in learning from numerical content. We constructed a Kolmogorov-Arnold Network (KAN) based Numerical Embedding Module and unified numerical and categorical features encoding within a Transformer architecture. TabKANet has demonstrated stable and significantly superior performance compared to Neural Networks (NNs) across multiple public datasets in binary classification, multi-class classification, and regression tasks. Its performance is comparable to or surpasses that of Gradient Boosted Decision Tree models (GBDTs). Our code is publicly available on GitHub: https://github.com/AI-thpremed/TabKANet.

## 1 INTRODUCTION

The tabular dataset is data organized into rows and columns, consisting of typically continuous, categorical, or ordered different features. It is the most commonly used and oldest data management model in building real-world businesses. At the same time, as the foundational data storage structure of relational databases, tabular data has widespread applications in almost any field, including medicineI (Johnson et al., 2016; Ulmer et al., 2020), finance (Clements et al., 2020; Arun et al., 2016), e-commerce (McMahan et al., 2013; Richardson et al., 2007), and more (Chandola et al., 2009; Buczak & Guven, 2015).

Despite recent advancements in neural network (NN) architectures for tabular data modeling, many still believe that the state-of-the-art techniques for performing tasks in tabular data, such as classification or regression, are based on tree ensemble methods, like Gradient Boosted Decision Trees (GBDT) (Hollmann et al., 2022; Chen & Guestrin, 2016; Prokhorenkova et al., 2018). This perspective is rooted in the observation that tree-based ensemble models often exhibit competitive predictive accuracy and much faster training speeds. This stands in stark contrast to widely researched fields of AI such as computer vision and natural language processing, where NN have significantly outperformed competing machine learning methods (Vaswani, 2017; Koroteev, 2021; OpenAI, 2023).

Although there is an ongoing debate about whether Neural Networks outperform GBDTs on tabular data (McElfresh et al., 2024). We must recognize that there are the distinct advantages of NNs in tabular data modeling:

- Potential for modeling large-scale complex tabular structures: Deep learning models may demonstrate superior performance.
- Better scalability at both the input and output ends: This provides a foundation for multi-modal applications.
- Self-supervised learning and pre-training schemes for tabular data offer greater potential for NNs in tabular modeling.

Therefore, modeling tabular data with neural networks is worth studying. There are attempts to utilize advanced neural networks like Transformer in tabular data modeling. More specifically, they have used Transformers to encode categorical items in tabular data (Huang et al., 2020), which we

believe is insufficient for they did not serve continuous numerical items in the same manner. In real-world datasets, numerical items are crucially important. If only the categorical parts are encoded using Transformers and continuous numerical items are simply fused through concatenation, it will lead to the unequal weighting of column items in model perception. We need a sensible tool to achieve vector mapping of continuous numerical items at a comparable scale while ensuring sufficient sensitivity to numerical distinctions.

Recently, the introduction of Kolmogorov-Arnold Network (KAN) (Liu et al., 2024b) has garnered significant attention from the AI communities, quickly piquing the interest of researchers and practitioners. A vital feature of the KAN is its ability to approximate functions of arbitrary complexity by selecting appropriate activation functions and parameters. This feature offers neural networks more flexible performance compared to Multilayer Perceptron (MLP) (Hornik et al., 1989). KAN shows a natural affinity of continuous numerical values, making it possible to become a significant component in processing the numerical features in tabular data.

In this study, we proposed the TabKANet, which conducted a targeted design for extracting numerical information from table data based on KAN, unifying category features and numerical features under the Transformers architecture. This is a new architecture designed for modeling tabular data, providing robust performance and a clear business structure framework.

We tested the performance of TabKANet, MLP (Hornik et al., 1989), KAN (Liu et al., 2024b), TabTransformer (Huang et al., 2020), TabNet (Arik & Pfister, 2021), XGBoost (Chen & Guestrin, 2016), and CatBoost (Prokhorenkova et al., 2018), across 12 widely used tabular tasks. TabKANet demonstrated consistent performance improvements across all tasks compared to other NN models, especially when the performance disparity between NNs and GBDTs is more pronounced. Our method achieved identical performance to the GBDT schemes in almost all datasets and even surpassed in multiple tasks. This indicates that our innovation in the KAN-based Numerical Embedding Module greatly exceeds earlier NN models and fully captures the potential value of numerical features in tabular data. It is worth noting that our work also emphasizes the combination of batch normalization with KAN has superior performance advantages in mapping multiple continuous numerical information.

## 2 RELATED WORK

### 2.1 TABULAR DATA MODELLING WITH NEURAL NETWORK

Tabular data is the primary format of data in real-world machine learning applications. Until recently, these applications were mainly addressed using classical decision tree models, such as GBDT. The XGBoost (Chen & Guestrin, 2016) and CatBoost (Prokhorenkova et al., 2018) are performance leaders in tabular data modeling for years.

However, with the development of deep neural networks, the performance gap between NNs and traditional GBDT models in table data tasks has begun to narrow. Recent advancements, exemplified by TabTransformer, have integrated transformers into tabular modeling approaches (Huang et al., 2020). TabNet employs a sequential attention mechanism to identify semantically significant feature subsets for processing at each decision point, drawing insights from decision tree methodologies (Arik & Pfister, 2021). TabPFN introduces a transformer architecture for in-context learning to approximate Bayesian inference through pre-training, facilitating swift resolutions for small tabular classification tasks without necessitating hyperparameter adjustments (Hollmann et al., 2022). FT-Transformer improved the ability of tabular data modeling by numerical embedding through linear transformation (Gorishniy et al., 2021). Recent research suggests that GBDTs perform better in handling datasets with skewed or heavy-tailed feature distributions than NNs, which may be due to their ability to better adapt to the irregularity of the data (McElfresh et al., 2024).

In addition, some studies explore the application of NNs in other areas of tabular data modeling, including few-shot learning, pretraining or imporve the performance by ensemble large language models (Hegselmann et al., 2023; Chen et al., 2023; Yin et al., 2020; Harari & Katz, 2022; Bertsimas et al., 2022; Nam et al., 2023; Ucar et al., 2021; Somepalli et al., 2021; Rubachev et al., 2022). There are also research in applying generative models for improving tabular task performance (Li et al., 2024b; Liu et al., 2024a; Kotelnikov et al., 2023).

## 2.2 KOLMOGOROV-ARNOLD NETWORK

MLPs have been the fundamental component of neural networks (Hornik et al., 1989). They feature a fully connected architecture that can approximate complex functions and possess expressive solid power, making them widely popular in various applications. However, despite their popularity, the MLP architecture also has some drawbacks. For instance, the activation functions are fixed. This rigidity in the network may limit the model's flexibility in capturing complex relationships within the data, as it relies on predefined nonlinear functions.

Liu et al. introduced the KAN as a alternative to MLPs to address the limitations (Liu et al., 2024b). Their research differs from previous studies because it recognizes the similarity between MLPs and networks that employ the Kolmogorov-Arnold theorem. A notable feature of KANs is the absence of traditional neural network weights. In KANs, each "weight" is represented as a small function. Unlike traditional NNs, where nodes apply fixed nonlinear activation functions, each edge in a KAN is characterized by a learnable activation function. This architectural paradigm allows KANs to be more flexible and adaptive than traditional methods, potentially enabling them to model complex relationships within the data.

With the introduction of KAN, researchers have been exploring their application to address scientific problems better, including time series forecasting, image segmentation, and hyperspectral image classification (Li et al., 2024a; Genet & Inzirillo, 2024; Lobanov et al., 2024). Previous work has also proposed to use the KAN network for tabular data modeling (Poeta et al., 2024). However, their method involves using the entire KAN model for table data classification and does not establish an excellent numerical feature embedding module, nor does it include Transformer architecture. This deficiency resulted in their work not fully demonstrating the potential of the KAN model in tabular data modeling.

## 3 THE TABKANET

For the vast majority of tables, the table data inevitably contains both continuous numerical and categorical items. In a table with $m$ categorical items and $n$ numerical items, we will handle these two types of data separately. Let $(x, y)$ denote a feature-target pair, where $x \equiv \{X_{\text{cat}}, X_{\text{num}}\}$. The $X_{\text{cat}} \in \mathbb{R}^m$ denotes all the categorical features and $X_{\text{num}} \in \mathbb{R}^n$ denotes all of the numerical features. Let $X_{\text{cat}} \equiv \{x_1, x_2, \dots, x_m\}$ with each $x_i$ being a categorical feature, for $i \in \{1, \dots, m\}$. Let $X_{\text{num}} \equiv \{x_1, x_2, \dots, x_n\}$ with each $x_i$ being a numerical feature, for $i \in \{1, \dots, n\}$. To achieve structural consistency of table data in neural networks, we need to first unify the dimensional expressions of numerical and categorical columns.

For category items, we encode all categories for each column, using methods such as One Hot Encoding or Label Encoding. For $X_{\text{cat}}$, the common practice is to map each categorical item to a specific dimensional space through an embedding layer. This results in a matrix that represents the categorical features:

$$f(\mathbf{X}_{\text{cat}}) = \begin{bmatrix} x_{11} & \cdots & x_{1d} \\ \vdots & \ddots & \vdots \\ x_{m1} & \cdots & x_{md} \end{bmatrix}$$

As mentioned in Sec.2.1, GBDTs outperform NNs in table modeling tasks because of the skewed or heavy-tailed features in table information. Current scientific research has not yet proposed a simple, stable, and universal numerical embedding module, which is an important bottleneck for NNs in table tasks (McElfresh et al., 2024). Therefore, improving the ability of NN in table modeling requires designing better solutions to achieve numerical embedding.

In a Kolmogorov-Arnold Network, the concepts of inner functions and outer functions are at the core of the network architecture:

Inner functions $\phi_{q,p}$ are univariate functions that operate on input variables $x_p$. They correspond to the first layer of the network, responsible for transforming each input variable into an intermediate representation. These functions deal with individual input features and output one or more values that will subsequently be used by the outer functions.

Outer functions $\Phi_q$ are functions that operate on the outputs of the inner functions. They correspond to the second layer of the network, responsible for summarizing and combining all intermediate values generated by the inner functions to produce the final output. These functions perform a weighted sum of the outputs of the inner functions to generate the final prediction results.

Mathematically, a multivariate continuous function $f$ can be represented as:

$$f(\mathbf{x}) = \sum_q \Phi_q \left( \sum_p \phi_{q,p}(x_p) \right)$$

Here, $\phi_{q,p}$ represents the inner functions, and $\Phi_q$ represents the outer functions.

In KAN, Basis splines (B-splines) are used to construct these inner and outer functions. B-splines are smooth functions defined piece-wise and are well-suited for approximating complex function shapes. By learning the control points of B-splines, KAN can adjust the shape of these functions to approximate the target function.

Inspired by the KAN Network, we have redesigned the numerical embedding module in table modeling. The architecture design of TabKANet is shown in Fig.1 and the illustration of the data flow procedure is shown in Fig.2.

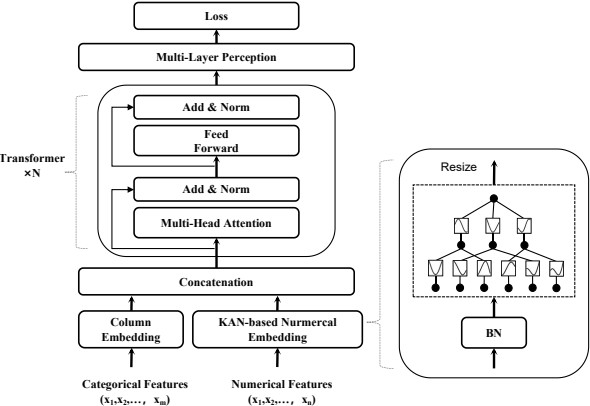

Figure 1: The architecture design of TabKANet.

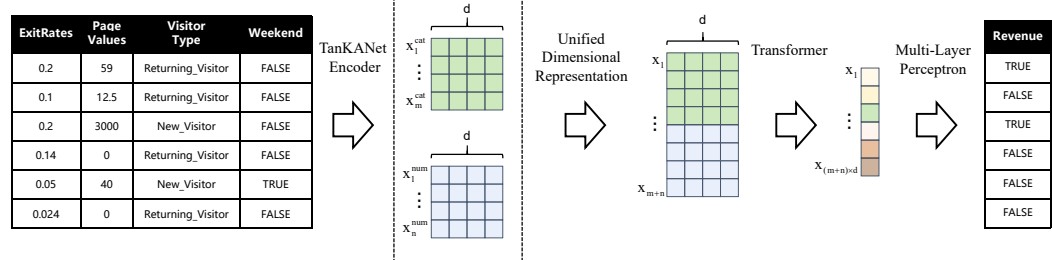

Figure 2: Illustration of data flow procedure in TabKANet. Implement dual-stream information extraction and achieve unified dimensional representation under Transformer architecture.

We have designed a KAN-based Nurmercal Embedding Module to better achieve feature extraction of numerical information. Specifically, we made the following three core improvements:

1. Replacing Layer Normalization (LN) in tabular data modeling for numerical items with Batch Normalization (BN).

2. Employing the KAN network for numerical items feature integration and extraction.

3. Mapping both categorical and numerical items into matrices of the unified dimension to feed into the Transformer, thereby leveraging the self-attention mechanism for modeling all data equally.

These improvements are based on the following considerations. Firstly, normalization for numerical items is crucial, which is essential to avoid gradient explosion, especially with real-world data. Previous work used LN to normalize numerical features (Huang et al., 2020). This is a subconscious best solution, which is to distribute numerical information as much as possible based on its global distribution. However, this may not be true. KAN also suffers from overfitting, and we need to ensure sufficient training during the learning process while ensuring authenticity to alleviate the skewed features. We have demonstrated the robust adaptability of BN through experiments in the 4.7 section, especially when combined with KAN, which can collect internal information on numerical values to a greater extent. As mentioned earlier, KAN, as a theoretically more powerful model than MLP, mainly when used on the input side of data, can extract information and stabilize changes better. In this process, BN and KAN work together, where BN is primarily used to alleviate data skew and amplify numerical differences, while KAN is responsible for integrating information, stabilizing output, and extracting internal value.

Additionally, since we cannot pre-determine the contribution weights of each column in tabular data, a pragmatic solution is to construct uniform representations for each column's features. This allows for the development of a unified downstream model for effective learning. Employing a consistent Transformer architecture for learning after encoding the column features also offers a highly scalable framework for tabular data modeling. Another important task is that when using a KAN-based Numerical Embedding Module, multiple training rounds can result in varying numerical normalization results due to the influence of BN. Repeatedly pairing numerical normalization results and category features will bring additional training data. We integrate these data features in Transformer, which is also important in significantly improving model performance.

## 4 EXPERIMENTS

### 4.1 DATASETS

In this study, we evaluated the TabKANet model and baseline models on 6 widely used binary classification datasets, 2 multi-class datasets, and 4 regression datasets from the UCI repository, OpenML, and Kaggle (Asuncion et al., 2007). These datasets exhibit extensive representativeness, spanning various domains, including finance, business, chemistry, geography, ecology, image recognition, and sports.

Table 1 details the characteristics of all datasets and their corresponding abbreviations used in this paper. All datasets are publicly available; the links can be accessed in the appendix in Table 10.

In our experimental design, each dataset is divided into five cross-validation folds, with the training, validation, and testing data ratio being 60%, 20%, and 20%, respectively. We utilize the Area Under the Curve (AUC) as the evaluation metric for binary tasks and the macro F1 score as the assessment metric for multi-class tasks. We employ the Root Mean Square Error (RMSE) for the evaluation metric of regression tasks.

### 4.2 BASELINE MODELS

As a comparison, we constructed the baseline MLP (Hornik et al., 1989) model, the baseline KAN (Liu et al., 2024b) model, the TabTransformer (Huang et al., 2020), the TabNet model (Arik & Pfister, 2021), the XGBoost (Chen & Guestrin, 2016), and the CatBoost model (Prokhorenkova et al., 2018). XGBoost and CatBoost are both gradient boost algorithms that utilize decision trees. TabNet is inspired by feature selection and combination strategies of GBDT and integrates these into a deep learning framework. MLP is a traditional deep learning model consisting of multiple layers of neurons. KAN model replaces MLP with Kolmogorov-Arnold Network to achieve table data learning. TabTransformer adapts the Transformer architecture for the categorical features of tabular data.

Table 1: The dataset feature used in this study.

| Dataset Name | Abbreviation | Num. Class | Num. Data | Num. Features |
|---|---|---|---|---|
| Blastchar Customer Churn | BL | 2 | 7,043 | 21 |
| Online Shoppers | ON | 2 | 12,330 | 18 |
| Seismic Bumps | SE | 2 | 2,583 | 19 |
| Biodegradation | BI | 2 | 1,055 | 42 |
| Credit Risk | CR | 2 | 1,000 | 21 |
| Bank Marketing | BA | 2 | 45,211 | 17 |
| Image Segmentation | SG | 7 | 2,310 | 20 |
| Forest Covertype | FO | 7 | 581,012 | 55 |
| CA House Prices | CA | - | 20,640 | 10 |
| Moneyball | MO | - | 1,232 | 15 |
| Sarcos Robotics | SA | - | 48,933 | 28 |
| CPU Predict | CPU | - | 8,192 | 13 |

## 4.3 RESULTS IN BINARY CLASSIFICATION

Table 2 shows the performance comparison between TabKANet and the comparison methods on 6 different datasets.

Table 2: Comparison between TabKANet and baseline NN methods. The evaluation metric is the mean $\pm$ standard deviation of AUC. The best performance is in **bold** for each row.

| Dataset | MLP | TabTransformer | KAN | TabNet | TabKANet |
|---|---|---|---|---|---|
| BI | 0.9033$\pm$0.035 | 0.9037$\pm$0.034 | 0.8987$\pm$0.021 | 0.8453$\pm$0.048 | **0.9110$\pm$0.032** |
| CR | 0.7468$\pm$0.037 | 0.7143$\pm$0.017 | 0.7193$\pm$0.048 | 0.7238$\pm$0.039 | **0.7727$\pm$0.047** |
| BL | 0.8276$\pm$0.014 | 0.8278$\pm$0.014 | 0.8259$\pm$0.015 | 0.8103$\pm$0.017 | **0.8284$\pm$0.013** |
| SE | 0.7314$\pm$0.032 | 0.7316$\pm$0.025 | 0.7189$\pm$0.026 | 0.7247$\pm$0.066 | **0.7495$\pm$0.043** |
| ON | 0.7229$\pm$0.012 | 0.7216$\pm$0.007 | 0.7785$\pm$0.029 | 0.9141$\pm$0.062 | **0.9195$\pm$0.005** |
| BA | 0.8873$\pm$0.002 | 0.8925$\pm$0.007 | 0.8966$\pm$0.003 | 0.9226$\pm$0.055 | **0.9321$\pm$0.004** |

TabKANet achieved the best performance compared to NN models across all datasets. In the ON dataset. TabKANet demonstrated a substantial improvement, achieving a 27.4% increase compared to TabTransformer. Our method greatly enhances the performance shortcomings of the NNs, especially for tabular datasets where the feature extraction capabilities of neural networks fall significantly behind GBDTs due to the limitations in numerical feature learning.

Table 3 shows the performance comparison between TabKANet and the GBDT methods in binary classification. In the CR and BL datasets, we have achieved performance advantages over GBDT. Meanwhile, the performance difference with the best GBDT model in all datasets is within 1.5%.

Table 3: Results for GBDTs and our proposed model in binary classification tasks. The evaluation metric is the mean $\pm$ standard deviation of AUC. The best performance is in **bold** for each row.

| Dataset | XGBoost | CatBoost | TabKANet |
|---|---|---|---|
| BI | 0.9187$\pm$0.032 | **0.9230$\pm$0.035** | 0.9110$\pm$0.032 |
| CR | 0.7686$\pm$0.041 | 0.7683$\pm$0.033 | **0.7727$\pm$0.047** |
| BL | 0.8137$\pm$0.013 | 0.8247$\pm$0.012 | **0.8284$\pm$0.013** |
| SE | 0.7556$\pm$0.027 | **0.7571$\pm$0.029** | 0.7495$\pm$0.043 |
| ON | 0.9272$\pm$0.006 | **0.9326$\pm$0.005** | 0.9195$\pm$0.005 |
| BA | 0.9298$\pm$0.002 | **0.9379$\pm$0.024** | 0.9321$\pm$0.004 |

## 4.4 RESULTS IN MULTI-CLASS CLASSIFICATION

Table 4 shows the performance comparison between TabKANet and the comparison NN models in 2 multi-class classification datasets. Since multi-class problems often involve challenges related to sample imbalance, we used macro F1 as the evaluation metric during training. The results indicate TabKANet achieved a significant performance advantage, showing substantial improvements in both macro F1 and macro ACC.

Table 4: Multi-class dataset using NN. The best performance is in bold for each row. Each score represents the average in 5-fold cross-validation. The best performance is in **bold** for each row. (The SG dataset only contains numerical terms and is unsuitable for TabTransformer.)

| Dataset | Metrics | Method | | | | |
|---------|---------|--------|----------------|--------|--------|----------|
| | | MLP | TabTransformer | KAN | TabNet | TabKANet |
| SG | ACC | 90.97 | - | 96.10 | 96.09 | **97.54** |
| | F1 | 90.73 | - | 95.69 | 94.96 | **97.56** |
| FO | ACC | 67.09 | 68.76 | 85.40 | 65.09 | **87.47** |
| | F1 | 48.03 | 49.47 | 71.56 | 52.52 | **72.76** |

Table 5 shows the performance comparison between TabKANet and the GBDT models in multi-class tasks. In the SG dataset, our performance difference from the best-performing XGBoost is within 0.5%, and in the FO dataset, we have achieved a leading advantage over GBDT in terms of both Macro F1 and Ac. The FO dataset has a huge amount of data, and experimental results demonstrate that TabKANet effectively utilizes the huge advantages of NN, demonstrating excellent performance in complex and larger data tasks. On the other hand, all methods achieved over 90% macro F1 in the SG dataset, indicating that this is a relatively simple task, and our proposed method still achieved a leading performance advantage.

Table 5: Results for GBDTs and our proposed model in multi-classification datasets. Each score represents the average in 5-fold cross-validation. The best performance is in **bold** for each row.

| Dataset | Metrics | Method | | |
|---------|---------|---------|----------|----------|
| | | XGBoost | CatBoost | TabKANet |
| SG | ACC | **97.90** | 97.39 | 97.54 |
| | F1 | **97.88** | 97.35 | 97.56 |
| FO | ACC | 86.03 | 82.13 | **87.47** |
| | F1 | 71.11 | 67.22 | **72.76** |

## 4.5 RESULTS IN REGRESSION TASKS

Table 6 shows the performance comparison between TabKANet and the comparison NN models in regression tasks. Table 7 shows the performance comparison of the top-performance NN models and the GBDT models in regression tasks. We used RMSE as an evaluation metric in regression tasks. TabKANet demonstrates a significant advantage over other NN models. Especially for the SA dataset with the largest data volume in the Regression datasets, TabKANet significantly outperforms all other methods. TabKANet achieved better results than TabNet and came very close to GBDTs for the CA dataset with a relatively large amount of data. On the other hand, for the MO dataset, which has a smaller number of data points (only 1,232), TabKANet appears to be less effective.

## 4.6 THE ROBUSTNESS TEST OF TABKANET

To demonstrate the superiority of TabKANet, we conducted further tests to evaluate its performance on noisy data and compared it with the performance of TabTransformer, baseline MLP, and KAN models. We performed the following experiments on the binary classification datasets ON, BA, and BL. On the test sample, we randomly replaced the categorical or numerical items every 10 percentage points, ranging from 10% to 50%. Then, input these test data into the trained model to predict their AUC scores.

Table 6: Comparison between TabKANet and baseline NN methods. The evaluation metric is the mean RMSE obtained from the 5-fold cross-validation testing procedure. The best performance is in **bold** for each row. (The SA and CPU datasets only contain numerical terms which are unsuitable for TabTransformer.)

| Dataset | MLP | TabTransformer | KAN | TabNet | TabKANet |
|---------|-----|----------------|-----|--------|----------|
| CA | 7.422 | 7.515 | 7.527 | 5.207 | **5.202** |
| MO | 0.799 | 0.801 | 0.779 | **0.339** | 0.386 |
| SA | 3.236 | - | 3.024 | 2.056 | **1.732** |
| CPU | 6.731 | - | 6.714 | 2.841 | **2.834** |

Table 7: Results for GBDT and top performance NN models in regression tasks. The evaluation metric is the mean RMSE obtained from the 5-fold cross-validation testing procedure. The best performance is in **bold** for each row.

| Dataset | XGBoost | CatBoost | TabNet | TabKANet |
|---------|---------|----------|--------|----------|
| CA | 4.731 | **4.462** | 5.207 | 5.202 |
| MO | 0.235 | **0.232** | 0.339 | 0.386 |
| SA | 2.173 | 2.297 | 2.056 | **1.732** |
| CPU | 2.926 | **2.683** | 2.841 | 2.834 |

In both the ON and BA datasets, TabKANet achieved a significant advantage over other deep learning models. As for the BL dataset, deep learning models even surpassed the capabilities of the GBDT method. However, the BL dataset has the fewest numerical features, with only 3 out of 20 input features.

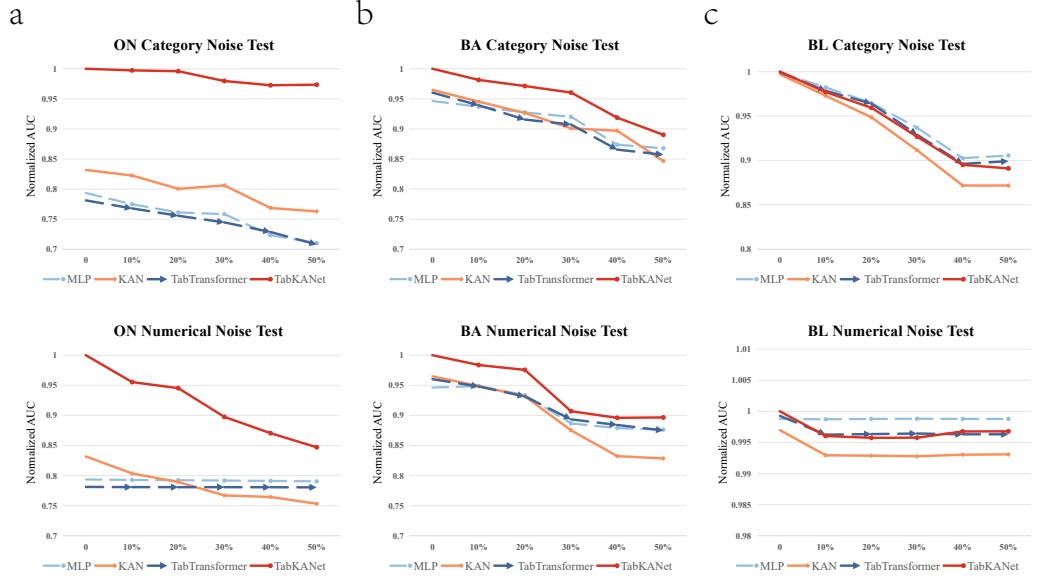

Figure 3: Robustness evaluation of TabKANet by introducing random noise for categorical and numerical features separately. a) Noise tests in the ON dataset. b) Noise tests in the BA dataset. c) Noise tests in the BL dataset.

Fig. 3a illustrates the predictive performance when random noise is introduced proportionally in the category features of the ON dataset. As the error rate of the category items increases, the performance of all models declines. However, TabKANet maintains a significant performance advantage and experiences only a 2.7% decrease in AUC when the category error rate reaches 50%. This result demonstrates that TabKANet exhibits stronger robustness as the error rate of category items

gradually increases. Similarly, as shown in Fig. 3b the BA dataset also shows similar results when categorical inputs contain errors.

It is particularly noteworthy that when the numerical entries in the ON dataset of Fig.3a exhibit a gradual increase in errors, the MLP and TabTransformer models are virtually unaffected. In contrast, TabKANet's performance gradually declines as the proportion of erroneous numerical entries increases. Eventually, when 50% of the numerical entries in the test set are incorrect, its performance advantage over TabTransformer diminishes from over 27% to just 8%. A similar trend is observed in Fig. 3b, as the proportion of erroneous numerical entries increases, the performance of all models declines, and TabKANet's performance converges with that of other deep learning models. This further demonstrates the performance advantage of TabKANet, which can significantly enhance the sensitivity to numerical features in the NN structure.

The ON dataset contains 6 numerical features, which other models have struggled to learn effectively. This inability to grasp the underlying information has led to a lack of response to errors in numerical error tests, a significant factor contributing to the underperformance of NNs compared to GBDTs. While the KAN model has shown some sensitivity, it outperforms MLP and TabTransformer in both the ON and BA datasets, this advantage is minimal and rapidly deteriorates with the introduction of numerical errors.

In Fig. 3c, we observe that as the categorical entries in the BL dataset become erroneous, all models experience a similar degree of performance decline. However, when numerical entries are corrupted, MLP remains largely unaffected, while the performance of the KAN model deteriorates more significantly than that of TabKANet and TabTransformer. As previously mentioned, deep learning methods generally outperform GBDT in the BL dataset. This indicates that numerical entries are not the performance bottleneck in the BL dataset, and our approach does not hinder the performance of deep learning models.

In summary, our method not only harnesses the enhanced extraction capabilities of KAN for numerical features but also, through a strategic combination, further strengthens this performance, significantly bolstering the overall stability of the model.

## 4.7 ABLATION EXPERIMENT

Unlike TabTransformer, this study also introduces the use of Batch Normalization instead of Layer Normalization for the extraction of numerical entries in tabular tasks (Huang et al., 2020). The rationale behind this approach is that BN can better capture the intrinsic feature differences within each numerical feature earlier in the learning process, thereby enhancing the model's ability to learn the skewed or heavy-tailed features in the numerical features.

Building on the same TabKANet model, Table 8 illustrates the performance differences between using LN and BN in binary classification tasks, with all models trained using a batch size of 128. TabKANet with LN shows improvement over TabTransformer, especially in tasks heavily reliant on numerical information such as BA and ON. However, when we employ BN for numerical normalization, the performance gain becomes even more pronounced.

For tasks where the improvement in numerical feature extraction does not yield significant benefits, such as the BL dataset, switching to BN for normalization does not diminish performance. This suggests that our approach is not only effective but also robust across various types of tabular data. For more experiments please refer to A.4.

Table 8: We compared the performance differences of TabTransformer using MLP to concatenate numerical items and the TabKANet using LN and BN. The best performance is in **bold** for each row.

| Dataset | TabTransformer | TabKANet-LN | TabKANet-BN |
|---------|----------------|-------------|-------------|
| BI | 0.9037±0.034 | 0.9048±0.032 | **0.9110±0.032** |
| CR | 0.7143±0.017 | 0.7495±0.042 | **0.7727±0.047** |
| BL | 0.8278±0.014 | 0.8283±0.012 | **0.8284±0.013** |
| SE | 0.7316±0.025 | 0.7375±0.016 | **0.7495±0.043** |
| ON | 0.7216±0.007 | 0.8841±0.009 | **0.9195±0.005** |
| BA | 0.8925±0.007 | 0.9197±0.003 | **0.9321±0.004** |

### 4.8 LIMITATION AND FUTURE WORKS

TabKANet has some potential limitations:

1. Since TabKANet's inclusion of the KAN network, Transformer, and MLP, despite achieving good results, it would perform even better if more detailed structural adjustments could be made to specific datasets or tasks.

2. Compared to traditional GBDT methods or GBDT-inspired methods (such as TabNet), table modeling developed based on NN, demand a significant increase in training time and hardware resources.

3. If some table tasks do not include numerical items or the contribution of numerical items is small, our solution may not be effective.

For future work, we believe it can focus on the following aspects:

1. This study is entirely based on supervised learning, which proves the performance of our proposed method in basic experiments. Semi-supervised and unsupervised pre-training methods have the potential to further improve its performance.

2. For real world tasks, especially artificial intelligence modeling of industrial processes involving large amounts of tabular data, it is necessary to provide guidance on model hyperparameters, structural scale size, and their relationship with tabular data through more extensive experiments.

3. TabKANet has shown performance in our experiments which is close to or even exceeds the GBDT methods, indicating the potential for building larger-scale table models for multimodal information.

## 5 CONCLUSIONS

In our study, we introduced TabKANet, a novel approach to table modeling that leverages a KAN-based numerical embedding module. The impressive performance of our model has been validated across a series of public tabular datasets, showcasing its advantages in terms of stability and ease of implementation. TabKANet's capability to effectively integrate information opens new probability for constructing intricate multimodal systems, potentially incorporating visual or language models. We are optimistic that TabKANet will serve as a solid foundation for future developments in table modeling, providing a versatile framework that can be expanded to address the challenges of tomorrow's data-driven landscape. Furthermore, the KAN-based numerical embedding module can be regarded as a flexible tool for enhancing the representation of numerical features in various applications.

### ACKNOWLEDGMENTS

Use unnumbered third level headings for the acknowledgments. All acknowledgments, including those to funding agencies, go at the end of the paper.

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

# A APPENDIX

## A.1 IMPLEMENTATION DETAILS

All categorical features in this study are encoded through Label Encoding. For MLP, KAN, Tab-Transformer, and TabKANet, we use the AdamW optimizer with a learning rate set to $1e-3$ for binary and multi-class classification tasks. We employ the SGD optimizer for regression tasks with

a learning rate of $1e - 4$. For TabNet, we utilize the pytorch_tabnet library with the Adam optimizer, setting the learning rate to $1e - 3$.

The XGBoost classifier is constructed using the XGBoost library in Python, with the maximum depth set to 8, the learning rate set to 0.1, and the number of trees set to 1000. The CatBoost classifier is built using the CatBoost library in Python, with the maximum depth set to 8, the learning rate set to 0.1, and the number of trees set to 1000.

All Transformer components are configured with the embedding dimension of 64, the number of heads to 8, and the number of layers to 3. The KAN network of TabKANet has one layer of neurons, with the number of neurons set to 64 and the embedding dimension also set to 64. For the output MLP, adjustments are made based on the total count of input categorical and numerical features.

## A.2 KAN NETWORK SETTING

In this paper, the number of numerical feature items in all datasets ranges from 3 to 27. In all our experiments with the KAN network, we have configured only one hidden layer, and the number of neurons in this hidden layer is fixed at 64. Furthermore, the number of output elements is set to $n \times 64$. To ensure rigor, we compared the scenario where the number of neurons was set to $2n + 1$ in the experiments with binary classification datasets, representing a dynamic allocation scheme for neurons based on the number of numerical inputs. Table 9 shows the performance differences between two different settings of the KAN module.

Overall, the difference caused by the two different settings is not significant. We suggest setting the number of hidden layer neurons in KAN to $\max(2n + 1, \dim)$ during the initial experiment. Here, $n$ represents the number of numerical input elements, and dim denotes the unified embedding dimension.

Table 9: Performance comparison between the dynamic allocation of neurons in KAN and the locked quantity setting. The best performance is in **bold** for each row.

| Dataset | TabKANet-dynamic | TabKANet-locked |
|---------|------------------|-----------------|
| BI | **0.9198$\pm$0.029** | 0.9110$\pm$0.032 |
| CR | 0.7600$\pm$0.038 | **0.7727$\pm$0.047** |
| BL | 0.8262$\pm$0.013 | **0.8284$\pm$0.013** |
| SE | **0.7632$\pm$0.037** | 0.7495$\pm$0.043 |
| ON | 0.9161$\pm$0.005 | **0.9195$\pm$0.005** |
| BA | 0.9294$\pm$0.004 | **0.9321$\pm$0.004** |

## A.3 DATASET INFORMATION

The links to the datasets used in this paper are shown in Table 10. We also display the number of numerical and categorical features in Table 11.

Table 10: Dataset links.

| Dataset | URL |
|---------|-----|
| Blastchar Customer Churn | https://www.kaggle.com/blastchar/telco-customer-churn |
| Online Shoppers | https://www.openml.org/search?type=data&status=any&id=45060 |
| Seismic Bumps | https://archive.ics.uci.edu/ml/datasets/seismic-bumps |
| Biodegradation | https://archive.ics.uci.edu/ml/datasets/QSAR+biodegradation |
| Credit Risk | https://www.openml.org/search?type=data&sort=runs&status=any&id=31 |
| Bank Marketing | https://archive.ics.uci.edu/ml/datasets/bank+marketing |
| Image Segmentation | https://www.openml.org/search?type=data&status=active&id=36 |
| Forest Covertype | https://www.openml.org/search?type=data&status=active&id=150 |
| CA House Prices | https://www.openml.org/search?type=data&status=active&id=43705 |
| Moneyball | https://www.openml.org/search?type=data&status=active&id=41021 |
| Sarcos Robotics | https://www.openml.org/search?type=data&status=active&id=44976 |
| CPU Predict | https://www.openml.org/search?type=data&status=active&id=227 |

Table 11: Number of categorical and numerical features in different datasets.

| Task | Dataset | #Cat. Features | #Num. Features |
|---|---|---|---|
| Binary Classfication | Blastchar Customer Churn | 17 | 3 |
| | Online Shoppers | 11 | 6 |
| | Seismic Bumps | 14 | 4 |
| | Biodegradation | 24 | 17 |
| | Credit Risk | 15 | 5 |
| | Bank Marketing | 10 | 6 |
| Multi-class | Image Segmentation | 0 | 19 |
| | Forest Covertype | 44 | 10 |
| Regression | CA House Prices | 1 | 8 |
| | Moneyball | 7 | 7 |
| | Sarcos Robotics | 0 | 27 |
| | CPU Predict | 0 | 12 |

## A.4 WHY IS BN BETTER

Our adoption of Batch Normalization for normalizing numerical features means that the choice of batch size during training can influence the model's performance. Fig. 4 illustrates the performance variation of TabKANet across three datasets when utilizing different batch sizes. The CR dataset contains the smallest amount of data, with only 1,000 data points, and its best-performing batch size is 64. Adjusting the batch size according to the total size of the training set when using Batch Normalization helps to improve the model's performance. In addition, this demonstrates the potential for further performance improvement by adjusting structural parameters based on TabKANet.

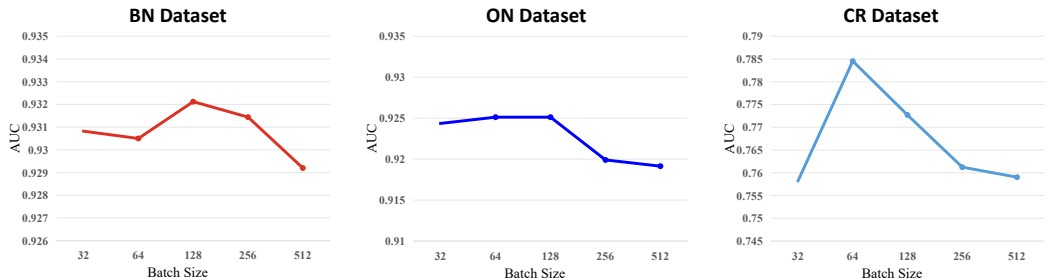

Figure 4: Impact of different batch sizes on TabKANet performance.

In Figure5, we employ the numerical term "Balance" from the BN dataset and "Informations_Duration" from the ON dataset to illustrate the variance in data interpretation between BN and LN. The horizontal axis represents the normalized ground truth of all numerical information, which is also the result obtained by LN. The vertical axis represents the normalized data values under different conditions. As the batch size increases, the results of BN and LN become increasingly similar.

The scatter plots reveal that employing BN with an optimal batch size can mitigate data skewness, amplifying subtle distinctions in the heavy-tail data. Concurrently, iterative numerical and categorical realignments were conducted throughout the training phase to bolster the model's capacity for generalization.

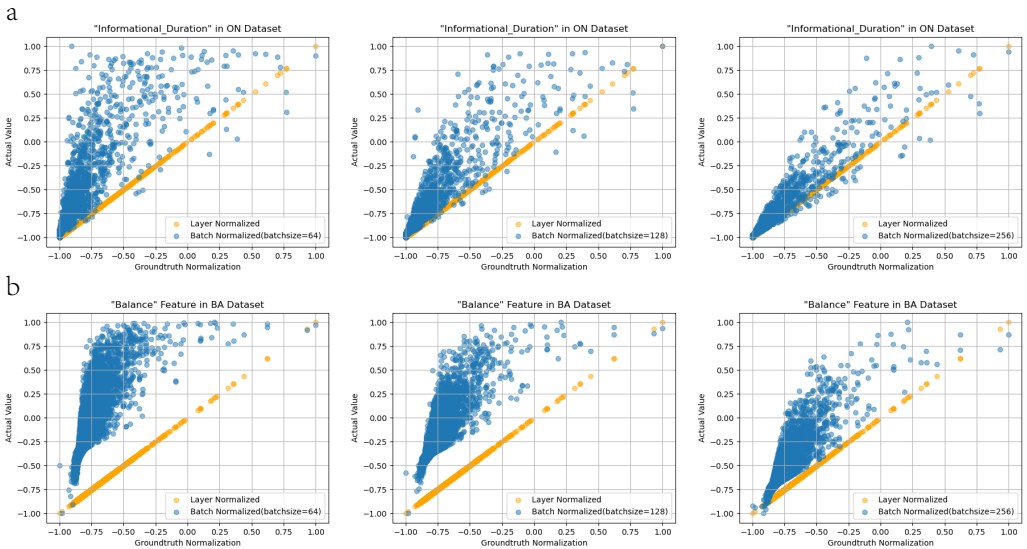

Figure 5: Display the effect of outputting two numerical items in two databases using Batch Normalization and Layer Normalization. a)"Informations_Duration" from the ON dataset. b)"Balance" from the BN dataset.

