# OpenReview forum: "TabKANet: Tabular Data Modeling with Kolmogorov-Arnold Network and Transformer"
_ICLR.cc/2025/Conference — ICLR 2025 Conference Withdrawn Submission_

### Official Review · Reviewer_xnza · 2024-10-29

**Soundness:** 2
**Presentation:** 2
**Contribution:** 2
**Rating:** 3
**Confidence:** 5

**Summary:**

The paper introduces TabKANet, a model that integrates Kolmogorov-Arnold Network (KAN) and Transformer architectures to improve the handling of structured tabular data. The authors demonstrate that TabKANet outperforms selected deep learning baselines in various tasks. However, its performance gains over decision tree-based ensemble methods, such as Gradient Boosted Decision Trees, are minimal.

**Strengths:**

- The paper addressing an important issue of the structured tabular learning
- Authors utilize various tabular datasets from different domains

**Weaknesses:**

- The explanation of KAN and the model’s overall structure is brief and lacks clarity. The paper’s organization could be improved to aid readers in understanding the background and methodology.

- The proposed method, TabKANet, offers only marginal improvements over existing models on key metrics for several datasets. For instance, on 2 out of 6 datasets, the AUC improvement compared to CatBoost is very small, with a difference of only **0.003**. This casts doubt on the claim from the abstract that *"Its performance is comparable to or surpasses that of Gradient Boosted Decision Tree models (GBDTs)."* Overall, such minor performance gains may not justify the added complexity of the approach.

- The paper omits several recently proposed methods for deep tabular learning, such as TabPFN, GANDALF,  DeepTLF, and more in [1]. Including these baselines would provide a more comprehensive comparison and better contextualize the performance of TabKANet.

- The evaluation of TabKANet could be strengthened by using widely recognized benchmarks, such as Tabzilla, which is well-accepted by the community and would provide a more robust assessment of the model's effectiveness.


[1] Borisov, Vadim, Tobias Leemann, Kathrin Seßler, Johannes Haug, Martin Pawelczyk, and Gjergji Kasneci. "Deep neural networks and tabular data: A survey." IEEE transactions on neural networks and learning systems (2022).

**Questions:**

- General question. Why are numerical and categorical data processed separately? Would it make sense to explore a combined representation that integrates both types of features?

- Line 221: The statement *"Firstly, normalization for numerical items is crucial, which is essential to avoid gradient explosion, especially with real-world data."* raises a question: Why not simply normalize the numerical data before feeding it to the neural network? Using batch normalization introduces additional learnable parameters and can significantly affect training, as the normalization depends on batch size. Although this question is somewhat addressed later (Line 238), it introduces another point of confusion: *"Repeatedly pairing numerical normalization results and category features will bring additional training data."* This may be more accurately described as data augmentation rather than additional data, which could be explored through an ablation study.

- Line 223: Could you clarify the phrase *"This is a subconscious best solution"*? The entire sentence is somewhat confusing and may need rephrasing for clarity.

- Line 255: The sentence about data splitting is unclear. Did you apply cross-validation, or was the data split into three groups: training, testing, and validation?

- Tables 2 and 3: What is the motivation for separating the neural network and machine learning baselines? A unified comparison would improve clarity.

Minor Comments

- In the motivation, you mention that self-supervised learning is a major strength of deep learning approaches. How could TabKANet be adapted for use in self-supervised settings?

- Section 3: The statement "As mentioned in Sec. 2.1, GBDTs outperform NNs in table modeling tasks because of the skewed or heavy-tailed features in table information" reads as a hypothesis rather than a definitive fact. Could this be clarified?

- Line 267: *"MLP is a traditional deep learning model consisting of multiple layers of neurons."* How are you defining a "traditional" deep learning model here? A more specific description might be helpful.

---

> ### Author Response · Authors · 2024-11-20
> **Response to review**
>
> On behalf of all the authors, I would like to express our sincere gratitude for your thorough review of our manuscript. Our goal was to develop a foundational model emphasizing simplicity. We kindly invite you to reconsider our response and re-evaluate the scores.
>
> Response to weakness:
>
> 1.	Thank you for your advice. We will add more information about KAN and its effectiveness in our model to make the paper clearer.
>
> 2.	GBDTs are strong methods that deep learning models strive to surpass. While conventional transformers have significantly lower performance, our TabKANet can surpass the state-of-the-art GBDTs, albeit with a minor lead.
>
> 3.	We completed this work prior to the publication of certain methods, so we missed including them in our study. We will try to add these methods for comparison if space permits.
>
> 4.	Thank you for the reminder. Although the number of datasets we used surpasses past research on tabular data modeling with transformers, we will also try to add more datasets to the article.
>
>
> Response to questions:
>
> 1.	According to past research, processing numerical and categorical data together often results in either low performance or low interpretability. It is a common and practical choice to fuse numerical and categorical information and process them separately for better performance.
>
> 2.	We will improve the clarity of “bring additional training data.” We are modifying it to read “training data will exhibit different values within different batches, which improves the generalization of subsequent modules.”
>
> 3.	We will rephrase this sentence, deleting “best subconscious” and other confusing terms.
>
> 4.	More descriptions of data segmentation will be updated in the article.
>
> 5.	This is a concern for us as well. Some research in tabular data modeling does not compare to ML methods because ML methods are too strong, making it difficult for their methods to outperform them. We aim to demonstrate that TabKANet is now strong enough to be compared to ML methods. Therefore, to emphasize this, we compare our method to both DL and ML methods separately.

---

> ### Comment · Reviewer_xnza · 2024-11-22
>
> >Thank you for the reminder. Although the number of datasets we used surpasses past research on tabular data modeling with transformers, we will also try to add more datasets to the article.
>
> I'm not suggesting bringing new datasets, rather than use acceptable-by-the-community benchmarks, see my review.
>
>
> Thank you for response, I do not see the new content, thus, I cannot change my score. Furthermore, some of my points were not addressed.

---

### Official Review · Reviewer_nyHP · 2024-11-01

**Soundness:** 2
**Presentation:** 2
**Contribution:** 2
**Rating:** 3
**Confidence:** 3

**Summary:**

The paper proposes a model for tabular data based on the Transformer architecture, similar to TabTransformer, but including a KAN layer for encoding continuous features. The model is evaluated on binary and multi-class classification tasks as well as regression and is found to outperform deep learning baselines and in some cases GBRT models.

**Strengths:**

The paper discusses encoding of continuous values for tabular classification. This is a hot topic, and the combination with the recently proposed KAN architecture is timely.
The model is evaluated on a wide variety of tasks and the datasets are discussed in detail.

**Weaknesses:**

# Main concern
Given that the novelty of the method is relatively small, the empirical evaluation of the method is critically important. However I have several concerns regarding this:
a) It's unclear how the datasets for evaluation were selected. There is many established benchmark suites, such as the AutoML benchmark, TabZilla, OpenML-CC18 and the Grinsztajn collection. Not adhering to a standard benchmark suite allows for cherry-picking of dataset.

b) Since the emphasis of the paper is on the encoding of continuous features, I think FTTransformer, and Gorishniy et al:  On Embeddings for Numerical Features in Tabular Deep Learning are critical baselines to compare against.

c) Some important ablations are missing; in particular, what happens if the KAN layer gets replaced by simple input scaling? This seems to be different form TabTransformer, which skips the transformer entirely for continuous data. Also, Table8 shows a big improvement between the LN and BN versions. An obvious comparison here would be to TabTransformer with BN.


d) The paper doesn't describe how hyper-parameters for the methods were tuned.

e) Some deep baselines are missing. While it's not reasonable to compare against all publications, I would suggest comparing against TabR and TabPFN (even when subsampled to a maximum of 3000 samples, TabPFN performance is still strong, see McElfresh.


I think the clarity of the paper could also be improved, and the novelty of the KAN is overstated, in particular wrt existing neural networks and "On Embeddings for Numerical Features".

# Other Concerns
## Overselling KAN
The paper states in several places that KANs are more powerful than MLPs; however, that is not strictly the case. A KAN can easily be represented with an MLP by constraining the MLP structure and potentially changing the activation function. For example a KAN with piecewise linear splines with r pieces is equivalent to an MLP with ReLU activations where each node is replaced by a small neural network with r nodes.  This view calls into question claims like line 63 "This feature offers neural networks
 more flexible performance compared to Multilayer Perceptron" and Line 115 "rigidity in MLP".
Also, neural networks with spline activation functions have long been studied, and it's unclear what (if any) novelty can be attributed to KANs.

## Minor suggestions
Line 028 "ordered different features" is hard to read and not very clear.

Line 028: add citations for most commonly used and oldest business data format. I am not convinced by these claims. A lot of data is actually in spreadsheets and relational databases, neither of which are tabular data in the sense of the standard ML datasets. NoSQL data is also extremely common.

Line 036: Citing Hollman for the prevalence of GRBT is strange, since TabPFN is a deep model that clearly outperforms GRBT.

Line 043: None of the three arguments for neural networks seems sound, and I am a firm believer in using neural networks on tabular data. 1) is vague, 2) it is unclear what is meant by scalability and how it relates to multimodality 3) unsupervised schemes exist for tree-based models, though they are not as common.

Line 070: It's unclear what is meant by "business structure framework"

Line 215 1): It's unclear wrt to what baselines you are discussing improvements. Both LN and BN are common in transformer models. Is this wrt KAN or wrt TabTransformer?

Line 218 3): This is just concatenating features, right?

Figure 2: The meaning of d is unclear, it seems to be embedding size. However, it's unclear why the output would be reshaped to (m+n) * d? Is this just to input into the MLP? Also, the figure shows multiple rows in input and output, but the model operates on one row at a time, right?

Line 223: It's unclear what's meant by "subconscious best solution".

Line 377: "predict auc scores" I think you mean predict the target and compute AUC scores?

## Typos

Line 027 "The tabular" -> "a tabular"

Line 030 "medicineI" ->"medicine"

Line 053 "they have used Transformers" -> Transformers have been used.

Figure 1: "Nurmerical" -> "Numerical"

Line 212: Numercal -> "Numerical"

Acknowledgements contain author instructions.

**Questions:**

* How were the datasets for the evaluation selected?
* How where hyper-parameters tuned for all models?
* How well does the model perform when removing the KAN layer?

---

> ### Author Response · Authors · 2024-11-20
> **Response to review**
>
> On behalf of all the authors, I would like to express our sincere gratitude for your thorough review of our manuscript. We kindly invite you to reconsider our response and re-evaluate the scores.
>
> Response to weakness:
>
> 1．	We have tested our model on more datasets than past deep learning tabular data modeling methods compared in our paper. While we consider the number of datasets one of our strengths, we acknowledge the need to add more datasets for validation.
>
> 2．	This method was replicated but not shown in our paper because it fails on several datasets due to gradient exploding. We identified that the lack of normalization in their model causes this issue, making it a defective model. Therefore, we did not include it in our paper.
>
> 3．	Comparisons with other encoders and workflows were conducted, and we selected this workflow with KAN for its best performance. These experiments may be updated in the article if space permits.
>
> 4．	Thank you for the reminder. We will consider adding this part to the ablation study.
>
> 5．	Thank you for your advice. We will add more comparison models.
>
> 6．	Model Clarity and Naming: We are working on improving clarity, which will be better after this revision. Your suggestions are valuable. We are also considering altering the model's name since KAN is not the only improvement in this model.
>
>
> Response to suggestions:
>
> 1.	The clarity and typo issues have been modified accordingly. Thank you again for your valuable suggestions.
>
> 2.	The datasets are mostly from past deep learning research papers; therefore, they are not typically "ML". We will try to add more comparison studies in our work.
>
> 3.	We are adding the comparison of TabPFN in the paper.
>
> 4.	This part will be modified due to its poor clarity. We aim to express that neural networks require almost constant inference time in practice, while GBDTs have higher inference time when the dataset is enormous.
>
> 5.	The rest of the clarity problems will be fixed. We really appreciate your patience and opinions.

---

### Official Review · Reviewer_dwTr · 2024-11-01

**Soundness:** 1
**Presentation:** 2
**Contribution:** 1
**Rating:** 3
**Confidence:** 4

**Summary:**

The paper introduces a discriminative model for tabular data that is distinct from previous models in that it uses a KAN for embedding numeric features rather than linear or MLP layers. Its performance is compared to a selection of GBDT and NN tabular models. In the experiment settings evaluated, TabKANet consistently outperforms the other NN models but is mostly outperformed by the GBDT models.

**Strengths:**

- Exploring the potential uses of KANs in tabular modelling is a relevant current topic.
- The model shows some promise in outperforming other NN models.
- Code is provided to reproduce some of the results.

**Weaknesses:**

On the whole, I think paper has a poor contribution due to issues with the evaluations and a lack of depth in justifying the proposed modelling choices. The overall novelty of the method is limited, being a relatively simple combination of existing modelling components (KANs, batch normalization, and transformers), and I don't think the rest of the paper has the depth or soundness to make up for that.

- No hyperparameter tuning is done for baseline models. This is especially problematic for GBDT models, which require hyperparameter tuning for a practical comparison. This is out of line with prior research (e.g., see Gorishniy et al. 2021) and severely limits the utility of the comparison with GBDTs.

- The NN models being compared to are not state-of-the-art, so the comparison to them does not indicate much about the paper's contribution. I would have at least liked to have seen TabR and TabPFN included, and other recent models that are widely used as baselines such as FT-Transformer and MLP-PLR would have been welcome.

- The discussion of why KANs should be used in this area is lacking in depth. The paper doesn't provide theoretical or empirical insights into how they would be useful for tabular data in particular. Instead, there are just vague references to their flexibility. Ablations are also not provided to compare the contribution of the KAN part alone versus other encoders.

- Using batch normalization instead of layer normalization is a fairly trivial tuning choice, and the discussion on page 5 is unclear and does not provide significant technical insights to justify treating it as an important decision.

- Parts of the paper contain misleading claims:
	- The introduction indicates that existing attempts to use transformers in tabular modelling are using them to encode categorical variables, when in fact there's a much wider variety of transformer models for tabular data (some of which are given in the Related Work). In general, the proposed model is not contextualized with respect to the entire range of existing tabular transformer models.
	- The introduction also claims that the proposed model achieved "identical performance" to GBDTs on almost all datasets - the performance was not identical, and was lower on average in most cases (evaluation issues notwithstanding).
	- "Current scientific research has not yet proposed a simple, stable, and universal numerical embedding module" - this is a very strong claim that is not justified. MLPs, linear layers, and piecewise linear encodings arguably satisfy these criteria just as well as the proposed solution.

**Questions:**

- What about KANs makes them a compelling choice for this application in particular: encoding numeric tabular data to pass into a downstream transformer? E.g., what theoretical properties are especially relevant for this application? Why not use them in other parts of the model?

- Did you evaluate other numeric encoders within the same framework as your model, such as linear, MLP, and/or piecewise linear encoders?

---

> ### Author Response · Authors · 2024-11-20
> **Response to review**
>
> On behalf of all the authors, I would like to express our sincere gratitude for your thorough review of our manuscript. Our goal was to develop a foundational model emphasizing simplicity. We kindly invite you to reconsider our response and re-evaluate the scores.
>
>
> Response to weakness:
>
> 1.	We have selected the best-performing GBDTs for comparison. Since our focus is on deep learning-based tabular data modeling, not all details are exhibited.
>
> 2.	Thank you for your advice. We will add more comparison models to our study.
>
> 3.	Encoder Selection: We conducted comparisons with other encoders and selected KAN for its best performance. These experiments may be updated in the article if space permits.
>
> 4.	We have discussed this on page 5 and conducted an ablation study in the later sections. Batch Normalization (BN) is very effective in this model, as shown in section 4.7. We aim to avoid excessive repetition to maintain conciseness.
>
> 5.	(1) The deficiencies of existing models in processing continuous values motivated this work. We do not imply that current models are only for categorical variable encoding but highlight their lack of exploration in continuous value processing. (2) We acknowledge that "resembling level of performance" would have been a more accurate expression, as most past transformer-based models perform lower than GBDTs. (3) The mentioned encoders are not as powerful as KAN. Again, comparisons with other encoders were conducted, and KAN was selected for its superior performance. These experiments may be updated in the article if space permits.
>
> Response to questions:
>
> 1.	Independent and separate processing of continuous and sparse values is a practical strategy in tabular modeling. We used KAN specifically for its effectiveness with continuous variables. After encoding, both types of variables are processed together by transformers, similar to most feature fusion methods and their downstream processes. We do not see this as problematic and are open to any advice you may have for improving our model.
>
> 2.	It seems this question is very important, as it has been addressed three times in this review. As mentioned, we conducted comparisons with other encoders and selected KAN for its superior performance. These experiments may be updated in the article if space permits.

---

> > ### Comment · Reviewer_dwTr · 2024-11-21
> >
> > Thank you for the response, I'll evaluate the additional paper content as it's added.
> >
> > To clarify the first point, I would expect hyperparameter tuning to be carried out for each dataset for baseline models. This could be skipped for NN models that don't typically use hyperparameter tuning, but I would certainly expect it for GBDTs and simple MLPs/KANs at least. In contrast, Appendix A.1 indicates that hyperparameters were fixed across all datasets for all models. I would again point to [1] as an example of appropriate hyperparameter tuning for fair comparisons.
> >
> > To clarify the intent behind my questions: I think the contributions of the paper would be more compelling if there were stronger theoretical and empirical justifications for using KANs as numeric input embedders for tabular models, given that embedding numeric inputs is a very common task, and simple, widely used and evaluated methods have enjoyed success. On the theoretical side, I think this needs to go beyond vaguely positive descriptions of KANs, such as calling them more powerful or adaptive, and into detailed reasoning on what specific features of the data and of KANs would make them effective (which would also answer the question of why to use them in just this specific part of the overall network). The original KAN paper does well on this front, justifying the performance of KANs in terms of specific mathematical results relating to compositional data structure, scaling laws, and the curse of dimensionality. On the empirical side, to show that KANs are driving improvements rather than some other modelling decision, it would be required to have head-to-head comparisons to other embedding approaches, which was why I was asking about those. A further direction you could consider taking the paper is using other non-Transformer backbones and seeing if KANs also help there, to show that the improvement isn't limited to your specific architecture. On the whole, I think the paper and the answers are indicating that KANs were just used because swapping MLPs out for them seemed to work better in this particular modelling setting, and that on its own without broader insights is not a deep contribution.
> >
> >
> > [1] Yury Gorishniy, Ivan Rubachev, Valentin Khrulkov, and Artem Babenko. Revisiting deep learning models for tabular data. NeurIPS 2021.

---

### Note · Authors · 2024-11-25

I have read and agree with the venue's withdrawal policy on behalf of myself and my co-authors.